depression; anxiety; mental health; medical student; Libya; storm Daniel

**Corresponding author:**
Mohammed S. Beshr;
Email: mbeshr2020@gmail.com

# The psychological impact of storm Daniel on medical students at the University of Derna in Libya: A cross-sectional study

Rana H. Shembesh[1] ⓘ, Mohammed S. Beshr[2] ⓘ, Aseel A. Almasheeti[1], Aisha T. Sheltami[1] and Ahmed El-Ojeli[3]

[1]Faculty of Medicine, Libyan International Medical University, Benghazi, Libya; [2]Faculty of Medicine and Health Sciences, Sana'a University, Sana'a, Yemen and [3]Community Medicine, Libyan International Medical University, Benghazi, Libya

## Abstract

Storm Daniel struck northeastern Libya on September 10, 2023, causing severe infrastructure damage and significant human loss. Derna was the most affected city, with the University of Derna suffering extensive damage and the tragic loss of 37 medical students. Medical students face unique psychological and academic stressors, and tend to have higher rates of psychiatric disorders compared to their peers of the same age. This is the first study to investigate the storm's psychological impact on medical students at the University of Derna. The study has a cross-sectional design and lasted from February 1 to March 1, 2024. We used the Generalized Anxiety Disorder-7 (GAD-7) to assess anxiety and the Patient Health Questionnaire-9 (PHQ-9) to assess depression, along with sociodemographic questions in our questionnaire. We included only active students enrolled in the 7-year undergraduate program at the University of Derna. Statistical tests such as the chi-square test and binary logistic regression were used in the analysis. About 225 students completed the survey. The means and standard deviations for GAD-7 and PHQ-9 scores were 9.2 (3.9) and 10.8 (5.0), respectively. The prevalence of anxiety was 42.2% for cases classified as moderate and severe (cut-off ≥10). Depression had a prevalence of 51.1% for cases classified as moderate, moderately severe and severe (cut-off ≥ 10). Suicidal ideation was reported at a rate of 48.9% for "several days" or more and at 16.5% for "more than half of the days" and "nearly every day." Internal displacement following the storm was significantly associated with both anxiety ($p = 0.033$) and depression ($p = 0.003$). However, age, gender, year of study, monthly allowance and residence status (living with family or alone) did not show a statistically significant association with either anxiety or depression ($p > 0.05$ for all variables). Logistic regression analysis identified gender as the only significant predictor of anxiety ($p = 0.041$) and internal displacement as the sole significant predictor of depression ($p = 0.023$). Medical students at the University of Derna reported high rates of anxiety, depression and suicidal ideation following Storm Daniel. Internal displacement was significantly associated with both anxiety and depression. These results highlight the need for targeted interventions to address medical students' mental health challenges and improve their overall well-being.

## Impact statement

Our evaluation of the impact of Storm Daniel on medical students at the University of Derna revealed higher levels of depression, anxiety and suicidal ideation among them. Natural disasters can have long-lasting negative mental health effects on survivors. There is a lack of mental health-related services in Libya. This study underscores the importance of developing a network of support and resources for the surviving medical students to help them cope, receive early diagnosis and get treatment if needed, with the goal of enhancing their mental health and overall well-being.

## Introduction

Natural disasters are an unavoidable global challenge. In addition to human loss and suffering, disasters negatively impact the mental health of individuals and communities (Kreimer 2001; Palinkas and Wong 2020). Survivors face an increased risk of severe post-traumatic psychopathologies, particularly depression and anxiety (Newnham et al. 2022). These effects are often more pronounced in lower-income countries (Davidson and McFarlane 2006).

On September 10, 2023, Storm Daniel struck northeastern Libya, bringing high winds and heavy rain that caused catastrophic floods and widespread destruction. Derna was the hardest-hit city, where two dams ruptured, sweeping away entire communities and destroying over 2,200 structures (Normand and Heggy 2024). The disaster led to 4,265 confirmed deaths, more than 8,500

missing persons, and nearly 45,000 displaced (WHO 2023). The University of Derna, a key institution in the city, was severely affected, with the tragic loss of 37 medical students, according to an official statement from the university.

This disaster unfolded amid Libya's ongoing recovery from the aftermath of a prolonged armed conflict, which devastated its infrastructure and healthcare system (Daw 2017; Fitzgerald 2023). A ceasefire agreed upon in October 2020 remains in place (Wikipedia 2024). Before the storm, a 2023 UNICEF report noted that over 525,000 people required health assistance and highlighted a critical lack of mental health services (UNICEF 2023). Derna and other eastern cities were already operating under suboptimal conditions, with only 10 semi- or fully functional healthcare facilities and multiple outpatient clinics (Fitzgerald 2023).

Within this challenging context, medical students are a unique demographic, particularly vulnerable to psychological stressors. A longitudinal cohort study assessed medical students before they entered medical school and followed them through their second and fourth years (Rosal et al. 1997). Before medical school, their mental health was similar to that of the general population, but after enrollment, depression rates increased and persisted. Compared to their peers in other disciplines, they face distinct academic and psychological challenges that may contribute to higher rates of mental health issues (Wolf et al. 1988). Globally, the prevalence of anxiety, depression and suicidal ideation among medical students is 33.8%, 27.2% and 11.1%, respectively (Rotenstein et al. 2016; Quek et al. 2019).

To date, this is the first study to assess the psychological impact of Storm Daniel on medical students in Libya. Our study focused on Derna University in Derna – the city most severely affected by the storm. There is a lack of studies focusing on the mental health of Libyan medical students, who face unique challenges due to ongoing political conflict. The ongoing uncertainty and instability continue to threaten their education and future.

We selected the Generalized Anxiety Disorder-7 (GAD-7) and the Patient Health Questionnaire-9 (PHQ-9) as our primary tools based on their well-established validity, standardized administration, capacity to classify symptom severity and widespread use across diverse populations (Spitzer et al. 2006; Löwe et al. 2008; Levis et al. 2019). Both are self-administered instruments, making them suitable for rapid assessment in a resource-limited post-disaster setting.

We included medical students from the University of Derna who were directly affected by the storm. Our aim was to assess whether there has been an increase in mental health disorders such as anxiety, depression and suicidal ideation among them.

## Materials and methods

### Study design, participants and settings

A cross-sectional design was conducted among medical students at the University of Derna following Storm Daniel. Data were collected from February 1 to March 1, 2024. We included only students who were actively attending the 7-year undergraduate program at the University of Derna, Faculty of Medicine, from the pre-med preparation year until the seventh internship year.

Students were asked to participate in the survey voluntarily. We used Google Forms to deliver our questionnaire electronically. All questions were presented without a skip option, ensuring that no responses were left unanswered or missing. The Libyan Red Crescent response team helped with messaging and encouraged

students to participate in the survey. For those who did not have internet access, the Red Crescent team provided access to ensure more participation in the survey.

We followed the STrengthening the Reporting of OBservational studies in Epidemiology Statement (STROBE), which outlines a checklist of items that should be included in reports of observational studies (von Elm et al. 2007). The STROBE checklist of this paper is available in the Supplementary Materials.

### Sample size

The sample size was calculated based on the following assumptions: the total number of medical students attending Derna University was 517, with an expected frequency of 50%, a margin of error of 5% and a 95% confidence interval. We chose 50% as the expected frequency. At the same time, we could have used the global anxiety rate among medical students (33.8%) (Quek et al. 2019). However, given the unusual circumstances surrounding medical students, such as exposure to the storm, we opted for 50% as it is more conservative and represents the highest possible variability in the population (Martínez-Mesa et al. 2014). The minimum required sample size was 221.

### Study tool

Our questionnaire was divided into three parts. The first part contained sociodemographic questions, including age, gender, year of study, monthly allowance, living status and internal displacement.

In the second section, we used the GAD-7, a tool developed to screen and evaluate the severity of anxiety disorders (Spitzer et al. 2006; Kroenke et al. 2007). This tool has been validated for screening anxiety disorders, with 82% specificity and 89% sensitivity (Spitzer et al. 2006; Löwe et al. 2008). Each item is rated on a 4-point scale: 0 (not at all), 1 (several days), 2 (more than half the days) and 3 (nearly every day), with scores ranging from 0 to 21 (Spitzer et al. 2006; Kroenke et al. 2007). The combined score is interpreted as follows: 0–4 (normal), 5–9 (mild), 10–14 (moderate) and 15–21 (severe) anxiety levels (Spitzer et al. 2006).

In the third section, we used PHQ-9. This validated nine-item tool is used to screen and assess an individual's level of depression (Spitzer et al. 1999; Kroenke et al. 2001). It has an 88% sensitivity and an 85% specificity (Levis et al. 2019). A 4-point scale for each item assessing depressive symptoms: 0 (absent), 1 (present for several days), 2 (present for more than half the days) and 3 (present nearly every day). A combined score ranging from 0 to 27 (Kroenke et al. 2001). Severity was then categorized as follows: 0–4 (minimal), 5–9 (mild), 10–14 (moderate), 15–19 (moderately severe) and 20–27 (severe) (Urtasun et al. 2019).

We used a validated Arabic translation of the PHQ-9 and GAD-7, previously tested on 731 university students, with Cronbach's alpha coefficients of 0.857 for the PHQ-9 and 0.763 for the GAD-7 (AlHadi et al. 2017). We conducted a pilot survey with 25 medical students and found that Cronbach's alpha was 0.83 for the PHQ-9 and 0.84 for the GAD-7, indicating a high level of internal consistency. The complete list of the questions used in our survey is available in the Supplementary Materials.

### Ethical considerations

The study protocol was approved by the Research Ethics Committee of the Libyan International Medical University in Benghazi,

Libya (Project Number: MHS-2-O-00197; Certificate Reference Number: MDC-2024-00160). The research adhered to the ethical principles of the Declaration of Helsinki, ensuring the protection and well-being of all participants.

A detailed information sheet was provided before participation, outlining the study's purpose, the voluntary nature of participation and confidentiality and anonymity measures. Participants were informed that they could withdraw at any time they wish. Only those who electronically confirmed their understanding and provided consent were granted access to the survey.

Recognizing the potential emotional distress associated with reflecting on recent traumatic experiences, psychological support was made available to participants. Consultant psychologists at Al-Ruwaie Psychiatric Hospital in Benghazi volunteered to provide assistance. The Red Crescent team communicated information about the availability of this support and provided guidance on how to access it.

Confidentiality was strictly maintained. Responses were collected anonymously, with no personally identifiable information recorded. Each participant was assigned a unique, randomly generated ID used solely for internal tracking. All data were securely stored on a password-protected account accessible only to the authors.

### Data analysis

We used IBM SPSS 29 version for the analysis. Both descriptive and inferential statistics were used in the analyses. Study variables were summarized using descriptive statistics, including frequencies, percentages, medians, interquartile ranges (IQRs), means and standard deviations. A univariate analysis using the chi-square test was performed to assess the associations between anxiety and depression and other variables. In addition, a binary logistic regression analysis was conducted to evaluate the effects of age, gender, year of study, monthly allowance, residence status and displacement on the likelihood of participants experiencing anxiety, and similarly, depression. A $p$-value of less than 0.05 was considered statistically significant.

### Results

### Characteristics of study subjects

The Faculty of Medicine at the University of Derna had a total of 517 students. Of these, 225 participated in the study. Among them, 118 (52.4%) were female and 107 (47.6%) were male. The median age was 23 (IQR, 21–26). The age distribution was as follows: 26.6% were between 18 and 21 years, 42.8% between 22 and 25 years, 23.6% between 26 and 29 years and 16% between 30 and 33 years. Regarding monthly allowance, 50.2% of individuals received 0–75 Libyan dinars, 32% received had 76–150 dinars and 17.8% received more than 150 dinars. Finally, approximately 34.2% reported being internally displaced following the storm. A detailed representation of the background characteristics is presented in Table 1.

### Anxiety

GAD-7 scores ranged from 0 to 21, with a mean and standard deviation of 9.23 (3.9). Based on these scores, anxiety severity was classified as follows: 50.7% of participants had mild anxiety (scores: 5–9), 33.3% had moderate anxiety (scores: 10–14) and 8.9% had

**Table 1.** Sociodemographic characteristics of the participants, *N* (%)

| Variable | *N* (%) |
|---|---|
| Total | 225 |
| Age | |
| 18–21 | 60 (26.6) |
| 22–25 | 96 (42.8) |
| 26–29 | 53 (23.6) |
| 30–33 | 16 (7) |
| Median (IQ) | 23 (21–26) |
| Gender | |
| Female | 118 (52.4) |
| Male | 107 (47.6) |
| Year of study | |
| Preparatory year | 14 (6.2) |
| First year | 45 (20) |
| Second year | 36 (16) |
| Third year | 41 (18.2) |
| Fourth year | 36 (16) |
| Fifth year | 33 (14.7) |
| Internship | 20 (8.9) |
| Personal monthly allowance | |
| 0–75 LYD | 113 (50.2) |
| 76–150 LYD | 72 (32) |
| More than 150 LYD | 40 (17.8) |
| Residence status | |
| With family | 183 (81.3) |
| Home alone | 42 (18.7) |
| Internal displacement | |
| Yes | 77 (34.2) |
| No | 148 (65.8) |

severe anxiety (scores: 15–21). A detailed classification of anxiety levels is presented in Table 2 and summarized in Figure 1.

In a univariate analysis, we assessed the association between anxiety symptoms (defined as a score ≥10) and various factors using the chi-square test. Age, gender, current year of study and residence status (with family or alone) did not show statistically significant associations, as all had a $p$-value greater than 0.05. However, monthly allowance was significantly associated with the presence of anxiety symptoms. A chi-square test yielded $\chi^2$ (2, $N = 225$) = 6.417, $p = 0.04$, indicating statistical significance. Similarly, internal displacement was also associated with the presence of anxiety symptoms, with the following values: $\chi^2$ (1, $N = 225$) of 4.539, and a $p$-value of 0.033, indicating statistical significance. A detailed representation of the associations with anxiety is shown in Table 3.

A multivariate logistic regression analysis was conducted to examine whether age, gender, year of study, monthly allowance, residence status and displacement influenced the likelihood of participants experiencing anxiety. The logistic regression model

**Table 2.** Classification of anxiety and depression based on GAD-7 and PHQ-9 scores

|  | Grade | N | Percentage |
|---|---|---|---|
| Anxiety (GAD–7) | Normal (0–4) | 16 | 7.1 |
|  | Mild (5–9) | 114 | 50.7 |
|  | Moderate (10–14) | 75 | 33.3 |
|  | Severe (15–21) | 20 | 8.9 |
| Depression (PHQ–9) | Minimal (0–4) | 13 | 5.8 |
|  | Mild (5–9) | 97 | 43.1 |
|  | Moderate (10–14) | 67 | 29.8 |
|  | Moderately severe (15–19) | 34 | 15.1 |
|  | Severe (20–27) | 14 | 6.2 |

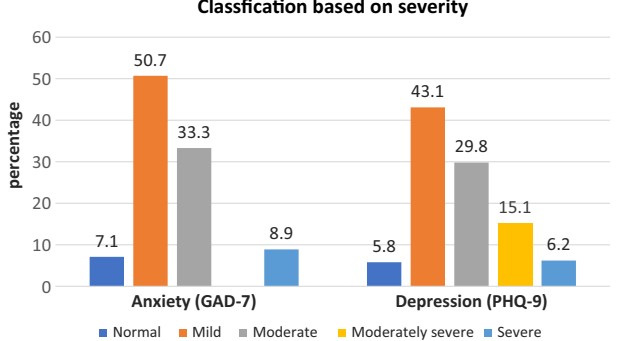

**Figure 1.** Classification of depression and anxiety based on the severity.

was statistically significant, $\chi^2$ (12, $N = 225$) = 22.335, $p = 0.034$. The model explained between 9.4% (Cox and Snell $R^2$) and 12.7% (Nagelkerke $R^2$) of the variance in the likelihood of anxiety. It correctly classified 63.1% of cases, with a sensitivity of 41.1% and specificity of 79.2%. Among the predictor variables, we analyzed, only gender was statistically significant (see Table 4). Being female was associated with a 1.978-fold increase in the likelihood of anxiety (OR: 1.978; 95% CI: 1.029–3.803; $p = 0.041$).

### Depression

The PHQ-9 scores ranged from 0 to 27, with a mean of 10.76 and a standard deviation of 5.02. Based on the classification of depression severity, 43.1% of participants had mild depression (scores: 5–9), 29.8% had moderate depression (scores: 10–14), 15.1% had moderately severe depression (scores: 15–19) and 6.2% had severe depression (scores: 20–27).

In a univariate analysis, depressive symptoms (defined as a score ≥10) were not significantly associated with age, gender, monthly allowance or residence status (living with family or alone); all had a $p$-value > 0.05. However, year of study was significantly associated with the presence of depressive symptoms and had a $\chi^2$ (6, $N$ = 255) of 13.394, and a $p$-value of 0.037, indicating statistical significance. Similarly, displacement after the storm was associated with the presence of depressive symptoms with the following metrics: $\chi^2$ (1, $N$ = 225) of 8.953, and a $p$-value of 0.003, indicating statistical significance. These associations are presented in detail in Table 5.

A multivariate logistic regression analysis was conducted to assess whether age, gender, year of study, monthly allowance, residence status or displacement had an influence on the likelihood of participants experiencing depression. The logistic regression model was statistically significant, $\chi^2$ (12, $N$ = 225) = 24.081,

**Table 3.** Univariate analysis of medical students' anxiety after Storm Daniel

| Variable | N | No anxiety symptoms PHQ-9 <10 | Anxiety symptoms PHQ-9 ≥10 | $\chi^2$ | $p$-value¶ |
|---|---|---|---|---|---|
| Total | 225 | 130 (57.8) | 95 (42.2) |  |  |
| Age |  |  |  | 4.501 | 0.212 |
| 18–21 | 60 (26.6) | 28 (21.5) | 32 (33.7) |  |  |
| 22–25 | 96 (42.8) | 61 (46.9) | 35 (36.8) |  |  |
| 26–29 | 53 (23.6) | 31 (23.8) | 22 (23.2) |  |  |
| 30–33 | 16 (7) | 10 (7.7) | 6 (6.3) |  |  |
| Gender |  |  |  | 2.788 | 0.095 |
| Female | 118 (52.4) | 62 (47.7) | 56 (58.9) |  |  |
| Male | 107 (47.6) | 68 (52.3) | 39 (41.1) |  |  |
| Year of study |  |  |  | 6.633 | 0.356 |
| Preparatory year | 14 (6.2) | 5 (3.8) | 9 (9.5) |  |  |
| First year | 45 (20) | 24 (18.5) | 21 (22.1) |  |  |
| Second year | 36 (16) | 20 (15.4) | 16 (16.8) |  |  |
| Third year | 41 (18.2) | 27 (20.8) | 14 (14.7) |  |  |
| Fourth year | 36 (16) | 23 (17.7) | 13 (13.7) |  |  |

*(Continued)*

**Table 3.** (*Continued*)

| Variable | N | No anxiety symptoms PHQ-9 <10 | Anxiety symptoms PHQ-9 ≥10 | $\chi^2$ | *p*-value[¶] |
|---|---|---|---|---|---|
| Fifth year | 33 (14.7) | 17 (13.1) | 16 (16.8) | | |
| Internship | 20 (8.9) | 14 (10.8) | 6 (6.3) | | |
| Personal monthly allowance | | | | 6.417 | **0.040** |
| 0–75 LYD | 113 (50.2) | 61 (46.9%) | 52 (54.7) | | |
| 76–150 LYD | 72 (32) | 50 (38.5%) | 22 (23.2) | | |
| More than 150 LYD | 40 (17.8) | 19 (14.6%) | 21 (22.1) | | |
| Residence status | | | | 0.193 | 0.661 |
| With family | 183 (81.3) | 107 (82.3) | 76 (80.0) | | |
| Home alone | 42 (18.7) | 23 (17.7) | 19 (20.0) | | |
| Internal displacement | | | | 4.539 | **0.033** |
| Yes | 77 (34.2) | 37 (28.5) | 40 (42.1) | | |
| No | 148 (65.8) | 93 (71.5) | 55 (57.9) | | |

*Note*: ¶, a *p*-value of less than 0.05 indicates statistical significance. Abbreviations: $\chi^2$, chi-square test. LYD, Libyan dinar.

**Table 4.** Logistic regression analysis of the effects of variables on the likelihood of anxiety

| Variable | Coef ($\beta$) | S.E. | OR | 95% CI | *p*-value[¶] |
|---|---|---|---|---|---|
| Age | 0.045 | 0.098 | 1.046 | 0.863–1.267 | 0.649 |
| Gender | 0.682 | 0.334 | 1.978 | 1.029–3.803 | **0.041** |
| Year of study | | | | | |
| Preparatory year (ref) | | | | | 0.360 |
| First year | −0.735 | 0.684 | 0.479 | 0.126–1.830 | 0.282 |
| Second year | −0.853 | 0.762 | 0.426 | 0.096–1.899 | 0.263 |
| Third year | −1.492 | 0.838 | 0.225 | 0.044–1.163 | 0.075 |
| Fourth year | −1.690 | 0.968 | 0.185 | 0.028–1.230 | 0.081 |
| Fifth year | −0.937 | 1.054 | 0.392 | 0.050–3.093 | 0.374 |
| Internship | −2.067 | 1.403 | 0.127 | 0.008–1.979 | 0.141 |
| Monthly allowance | | | | | |
| 0–75 LYD (ref) | | | | | 0.064 |
| 76–150 LYD | −0.452 | 0.343 | 0.637 | 0.325–1.246 | 0.187 |
| More than 150 LYD | 0.576 | 0.442 | 1.779 | 0.748–4.234 | 0.193 |
| Residence status | −0.295 | 0.457 | 0.744 | 0.304–1.823 | 0.518 |
| Internal displacement | 0.598 | 0.315 | 1.819 | 0.980–3.375 | 0.058 |
| Constant | −0.537 | 2.081 | 0.585 | | 0.796 |

*Note*: ¶, a *p*-value of less than 0.05 indicates statistical significance. Abbreviations: Coef ($\beta$), coefficient beta of the variables; S.E., standard error; OR, odds ratio; 95% CI, 95% confidence interval; ref, reference; LYD, Libyan dinar.

$p = 0.02$. The model explained between 10.1% (Cox and Snell $R^2$) and 13.5% (Nagelkerke $R^2$) of the variance in the likelihood of depression. It correctly classified 61.8% of cases, with a sensitivity of 60% and specificity of 63.6%. Among the predictor variables, only internal displacement was statistically significant (see Table 6). Being internally displaced increased the odds of depression by 2.051 (95% CI: 1.106–3.805; $p = 0.023$).

### Suicide ideation

In our assessment of suicidal ideation, 115 (51.1%) of students reported having no suicidal thoughts, 73 (32.4%) thought about suicide on several days, 29 (12.9%) experienced suicidal ideation more than half the days and 8 (3.6%) thought about it nearly every day. The results of suicidal ideation are presented in Figure 2.

### Discussion

Before the storm, Libya was still recovering its weakened healthcare system from past conflicts (Daw 2017; Fitzgerald 2023). A 2023 UNICEF report noted that over 525,000 people needed health assistance and highlighted a critical lack of mental health support services (UNICEF 2023). Eastern cities like Derna were already operating under suboptimal conditions, with only 10 healthcare facilities being semi- or fully functional (Fitzgerald 2023).

In our study, medical students at Derna University reported high rates of anxiety and depression. Anxiety was experienced by 42.2% of students, while 51.1% reported symptoms of depression. Displacement was significantly associated with both conditions. For suicidal ideation, 48.9% reported some level of suicidal thoughts, and 16.5% experienced them frequently.

Natural disasters have consistently been linked to increased psychological distress among college students. After the 2023 earthquake in Morocco, depression and anxiety were reported in 42.01% and 37.59% of nursing students, with higher rates among those evacuated or displaced (Achbani et al. 2024). Similarly, the 2023 Kahramanmaraş earthquake in Turkey was associated with elevated anxiety among students who directly experienced the event or suffered material losses (Kaya and Bayram 2024). Comparable mental health impacts have been documented worldwide: after the 2015 Nepal earthquake, 43.2% of college students reported depression (Sharma et al. 2021), while the 2010 Haiti earthquake saw high

**Table 5.** Univariate analysis of medical students' depression after Storm Daniel

| Variable | N | No depressive symptoms PHQ-9 <10 | Depressive symptoms PHQ-9 ≥10 | $\chi^2$ | *p*-value[¶] |
|---|---|---|---|---|---|
| Total | 225 | 110 (48.9) | 115 (51.1) | | |
| Age | | | | | |
| 18–21 | 60 (26.6) | 30 (27.3) | 30 (26.1) | 0.352 | 0.955 |
| 22–25 | 96 (42.8) | 46 (41.8) | 50 (43.5) | | |
| 26–29 | 53 (23.6) | 27 (24.5) | 26 (22.6) | | |
| 30–33 | 16 (7) | 7 (6.4) | 9 (7.8) | | |
| Gender | | | | 3.191 | 0.074 |
| Female | 118 (52.4) | 51 (46.4) | 67 (58.3) | | |
| Male | 107 (47.6) | 59 (53.6) | 48 (41.7) | | |
| Year of study | | | | 13.394 | **0.037** |
| Preparatory year | 14 (6.2) | 4 (3.6) | 10 (8.7) | | |
| First year | 45 (20) | 27 (24.5) | 18 (15.7) | | |
| Second year | 36 (16) | 19 (17.3) | 17 (14.8) | | |
| Third year | 41 (18.2) | 19 (17.3) | 22 (19.1) | | |
| Fourth year | 36 (16) | 10 (9.1) | 26 (22.6) | | |
| Fifth year | 33 (14.7) | 20 (18.2) | 13 (11.3) | | |
| Internship | 20 (8.9) | 11 (10) | 9 (7.8) | | |
| Personal monthly allowance | | | | 1.896 | 0.388 |
| 0–75 LYD | 113 (50.2) | 52 (47.3) | 61 (53) | | |
| 76–150 LYD | 72 (32) | 40 (36.4) | 32 (27.8) | | |
| More than 150 LYD | 40 (17.8) | 18 (16.4) | 22 (19.1) | | |
| Residence status | | | | 0.275 | 0.6 |
| With family | 183 (81.3) | 91 (82.7) | 92 (80) | | |
| Home alone | 42 (18.7) | 19 (17.3) | 23 (20) | | |
| Internal displacement | | | | 8.953 | **0.003** |
| Yes | 77 (34.2) | 27 (24.5) | 50 (43.5) | | |
| No | 148 (65.8) | 83 (75.5) | 65 (56.5) | | |

*Note*: ¶, a *p*-value of less than 0.05 indicates statistical significance. Abbreviations: $\chi^2$, chi-square test. LYD, Libyan dinar.

rates of PTSD (36%), depression (31.7%) and anxiety (21.1%) among university students (Silvestre et al. 2014). The 2022 floods in Pakistan also disrupted education, delaying school reopenings and causing trauma, depression, anxiety and fear of future disasters (Gul et al. 2024).

Similar trends have been observed in Middle Eastern countries affected by recent conflicts and unrest. For example, in Yemen, 48.4% of medical students experienced depressive symptoms, 34.8% had anxiety and 33.5% reported suicidal ideation (Beshr et al. 2024). Similarly, Al Saadi et al. (2017) found that 60.6% of Syrian medical students experienced depression and 35.1% had anxiety, using the DASS-21 scale. During the recent armed conflicts in Sudan, medical students reported experiencing high rates of depression (58.3%) and anxiety (51.5%). The ongoing conflict, instability and uncertainty continue to threaten their future, with 29.74% considering dropping out and 55.56% thinking about transferring (Alfadul et al. 2025). Our findings from Derna, particularly the elevated rates of anxiety, depression and suicidal ideation, align with the psychological distress reported among

medical students in other conflict-affected regions across the Middle East.

Libya's preexisting fragility, resulting from years of armed conflict, may have amplified the psychological impact of Storm Daniel on medical students. The prolonged conflict had already strained resources, destabilized communities and weakened the healthcare system. The cumulative burden of enduring years of instability – followed by a catastrophic natural disaster with limited support – likely contributed to the high prevalence of psychological distress observed in our study.

Psychological models of PTSD, such as the Cognitive Model of PTSD (Ehlers and Clark 2000) and the Dual Representation Theory (Brewin et al. 1996), describe how individuals process traumatic events. These frameworks may help explain the higher rates of anxiety, depression and suicidal ideation observed in our study. These models suggest that intrusive memories, maladaptive appraisals and disruptions in autobiographical memory consolidation can heighten emotional distress after a disaster. Given the extreme nature of Storm Daniel, medical students may have been

**Table 6.** Logistic regression analysis of the effects of variables on the likelihood of depression

| Variable | Coef (β) | S.E. | OR | 95% CI | *p*-value¶ |
|---|---|---|---|---|---|
| Age | −0.009 | 0.097 | 0.991 | 0.819–1.198 | 0.924 |
| Gender | 0.393 | 0.325 | 1.481 | 0.784–2.799 | 0.227 |
| Year of study | | | | | |
| Preparatory year (ref) | | | | | 0.138 |
| First year | −1.241 | 0.706 | 0.289 | 0.072–1.154 | 0.079 |
| Second year | −0.869 | 0.777 | 0.420 | 0.091–1.926 | 0.264 |
| Third year | −0.639 | 0.842 | 0.528 | 0.101–2.749 | 0.448 |
| Fourth year | 0.022 | 0.985 | 1.023 | 0.148–7.047 | 0.982 |
| Fifth year | −1.170 | 1.063 | 0.310 | 0.039–2.493 | 0.271 |
| Internship | −1.062 | 1.391 | 0.346 | 0.023–5.278 | 0.445 |
| Monthly allowance | | | | | |
| 0–75 LYD (ref) | | | | | 0.534 |
| 76–150 LYD | −0.318 | 0.336 | 0.728 | 0.377–1.405 | 0.344 |
| More than 150 LYD | 0.080 | 0.444 | 1.083 | 0.454–2.585 | 0.858 |
| Residence status | −0.342 | 0.454 | 0.710 | 0.292–1.728 | 0.451 |
| Internal displacement | 0.718 | 0.315 | 2.051 | 1.106–3.805 | **0.023** |
| Constant | 0.960 | 2.068 | 2.612 | | 0.642 |

*Note*: ¶, a *p*-value of less than 0.05 indicates statistical significance. Abbreviations: Coef (β), coefficient beta of the variables; S.E., standard error; OR, odds ratio; 95% CI, 95% confidence interval; ref, reference; LYD, Libyan dinar.

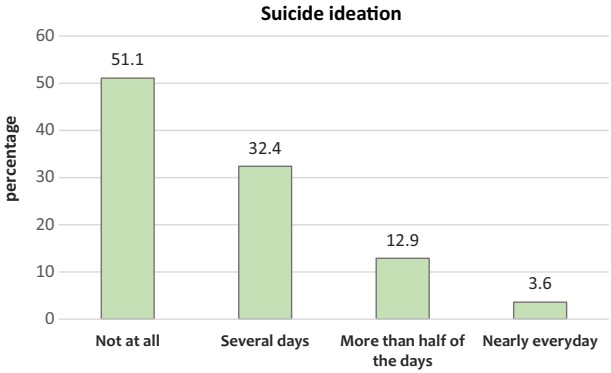

**Figure 2.** Prevalence of suicidal ideation among Derna medical students after Storm Daniel.

particularly vulnerable due to personal losses, displacement and academic disruptions. Frequent exposure to distressing scenes, loss of loved ones or property and academic disruption may have contributed to negative changes in cognition and mood, as reflected in our prevalence rates.

Crisis theory also provides a framework for understanding how individuals cope with sudden, overwhelming stressors (WHO 2023). Initially, individuals may experience distress, anxiety and confusion. They attempt to manage the situation using familiar coping strategies. If these fail, distress can intensify, leading to psychological dysfunction. Ultimately, the crisis may be resolved either positively – promoting growth and adaptation – or negatively, resulting in a decline in mental health. Adaptation depends on internal resilience, support systems and timely intervention, with earlier support generally associated with better outcomes (Wolkon 1972; Flannery and Everly 2000; WHO 2023). For Derna medical students, the combined stress of the storm and preexisting vulnerabilities from prolonged conflict might have overwhelmed their coping mechanisms. The absence of immediate, adequate mental health support – further strained by a weakened healthcare infrastructure – may have hindered their recovery.

In our evaluation of associations with anxiety and depression, several variables were significant. Displacement was significantly associated with anxiety and depression in our study. Over 25% of the buildings in Derna were destroyed (Eye ME 2024). About 34.2% of our participants were internally displaced, with over half of them experiencing symptoms of depression and anxiety. Students were forced to take shelter with relatives in nearby cities or to rent accommodations. Multiple studies confirmed that the risk of psychological distress is higher in those who are internally displaced (Davis et al. 2010; Morina et al. 2018; Rizzi et al. 2022). This may explain the significant associations of internal displacement with both depression and anxiety.

The year of study was significantly associated with depression, with the fourth year showing higher rates. This may be because in Libya, the fourth year marks the transition from the preclinical to the clinical phase of medical education. A study of U.S. medical schools revealed that clinical years carry a greater risk of depression compared to preclinical years (Chandavarkar et al. 2007).

Our study found a significant association between anxiety and monthly allowance, with 54.7% of those experiencing anxiety symptoms reporting "insufficient" allowance (<75 Libyan Dinar). Financial insecurity has been shown to negatively impact the mental health and academic performance of medical students (Ross et al. 2006). It contributes to chronic stress, uncertainty about the future, and limited access to care, exacerbating anxiety and depression (Jessop et al. 2005; Eisenberg et al. 2013). The financial burden may also prevent students from seeking professional help, further worsening their symptoms. To mitigate these barriers, institutions should provide free or low-cost counseling, telehealth services and peer-support networks. These support groups can reduce isolation and promote shared coping strategies.

The prevalence of suicidal ideation among medical students in our study was alarmingly high, with 48.9% reporting some level of suicidal thoughts and 16.5% experiencing them frequently. These rates far exceed those reported globally and regionally. For example, a systematic review on medical students by Rotenstein et al. reported a global prevalence of 11.1%, while studies in the Middle East and North Africa typically range from 10 to 26% (Amiri et al. 2013; Ahmed et al. 2016; Rotenstein et al. 2016; Madadin et al. 2021; Beshr et al. 2024; Dodin et al. 2024). Several factors may explain the elevated rates at the University of Derna. The traumatic impact of Storm Daniel, the loss of 37 peers, educational disruption and displacement – all of which may have contributed to psychological distress. Internal displacement may exacerbate this by disrupting stability, social support and coping mechanisms. Furthermore, cultural and systemic barriers to mental health care in Libya may worsen distress by limiting access to support and professional intervention.

In our regression analysis, although the models were statistically significant, the predictors accounted for only a modest proportion of the variance in anxiety and depression (Nagelkerke $R^2 \sim$ 10–13%),

suggesting that other unmeasured factors likely contribute to mental health outcomes. These may include personal loss, prior mental health history, social support, academic pressure, coping mechanisms, environmental stressors or trauma exposure. Future research should incorporate a broader range of psychosocial and contextual variables to better understand mental health in disaster settings. Nevertheless, our findings highlight key demographic predictors – gender for anxiety and displacement status for depression – that align with patterns in disaster-related mental health literature (Munro et al. 2017; McKinzie and Clay-Warner 2021). In the univariate analysis, gender was not significantly associated with anxiety. However, in the multivariate model – which adjusted for other predictors – the independent effect of gender became apparent and statistically significant.

Derna and other eastern cities were already operating under suboptimal conditions. After the storm, over 50% of these facilities became partially or completely nonfunctional, further hindering response and mitigation efforts (Fitzgerald 2023). UNICEF, the Red Crescent, foreign aid organizations and the Libyan government collaborated on infrastructure restoration in Derna, providing medical aid, food and shelter during the early stages of the crisis.

The University of Derna housed displaced families in the initial weeks and supported survivors with care and shelter. Following the tragic loss of 37 medical students, the university promptly acknowledged the event, issued a statement expressing condolences and solidarity with the affected families, and held a tribute in their memory. It also provided emotional and spiritual supports to help students cope, as many were experiencing shock and grief. In the aftermath, the university began rebuilding its facilities and officially resumed classes on October 2, 2023 – 1 month after the disaster. To this day, they are still dealing with the consequences of the storm.

After the storm, no mental health services were available in Derna. The only psychiatric counseling for medical students was offered by Al-Ruwaie Psychiatric Hospital in Benghazi, a city four-and-a-half-hour drive. Benghazi, which has relatively better healthcare facilities, hosted the majority of displaced people and most of the medical services to survivors. No on-campus mental health counseling was established for medical students.

Libya faces numerous challenges and barriers in providing mental health support services (Rhouma et al. 2016; UNICEF 2023). The healthcare system suffers from limited resources, damaged infrastructure and a shortage of qualified professionals, all of which hinder effective service delivery. Geographic barriers, such as vast distances and poor transportation, further restrict access, particularly in areas like Derna. In addition, political instability and security concerns continue to disrupt service planning and delivery. The lack of reliable data on the prevalence and outcomes of mental health issues impedes efficient resource allocation and program development. Moreover, stigma surrounding psychiatric and psychological disorders remains a substantial barrier in Libya. Unfortunately, cultural attitudes and beliefs often discourage individuals from seeking help.

To address these gaps and barriers, a comprehensive policy reform is urgently needed to strengthen Libya's health system and improve mental health services. Key priorities should include integrating mental health into primary care to improve accessibility. Training of mental health professionals is needed to expand mental health resources. Collaborating with local mental health organizations and policymakers could help increase counseling resources and establish long-term support systems. In addition, it is essential to ensure adequate funding and creating frameworks for

telehealth to reach underserved areas and improve access to mental health care.

For medical students, university-based programs such as counseling services tailored to students who have experienced trauma are needed. Peer support groups that provide a safe space for sharing experiences, and affordable telehealth services, can help reduce financial barriers to care. Additionally, resilience-building programs, such as stress management workshops and mindfulness training, may promote effective coping strategies. These efforts will enhance students' well-being and improve their academic performance.

Finally, integrating mental health education into the curriculum can foster awareness and reduce stigma, encouraging students to seek help when necessary. Nationwide efforts, including targeted awareness campaigns, are essential to normalize mental health care and promote help-seeking behaviors.

Conducting a survey in a post-disaster setting presents several ethical challenges. Although the risk of participation is minimal, the potential for emotional distress when reflecting on recent trauma must be acknowledged. Providing real-time psychological support or mental health resources is important. Ensuring participants feel safe sharing their experiences requires a thorough informed consent process, along with strict anonymity measures. These steps likely enhanced participant trust and encouraged honest responses on sensitive topics. However, we were unable to offer real-time psychological support or follow up with participants reporting severe distress due to the anonymous, online nature of the survey.

Future research should further explore the cumulative and long-term effects of conflict and natural disasters on mental health among medical students and other vulnerable groups. Longitudinal studies are needed to assess the persistence of psychological symptoms and to evaluate coping mechanisms, quality of life and academic performance among affected individuals. Moreover, research should assess the effectiveness of mental health interventions tailored to these complex environments. Finally, future research should help in gathering more data on mental health prevalence and outcomes, thereby enhancing resource allocation and program development.

## *Limitations*

Due to the cross-sectional design, we cannot establish causation from the findings. While our findings suggest a potential link between the storm and students' mental health outcomes, this design prevents us from establishing causation. We used a retrospective self-reporting survey, which might be vulnerable to recall bias. We did not evaluate the students' psychiatric history or the presence of any underlying mental illnesses, so we cannot draw associations. Factors such as the severity of exposure, bereavement (e.g., loss of family or friends), prior trauma and levels of social support were not examined in the survey but may influence psychological outcomes. Future research should incorporate these variables into its survey. Since this survey was voluntary, the results might be less representative due to a lack of participation. Although we provided a thorough informed consent form and ensured anonymity to build trust and encourage participation, some students may have opted out due to fear of emotional discomfort, despite assurances of confidentiality. This may have introduced a self-selection bias. Additionally, since the survey was conducted online, we may not have reached individuals from low socioeconomic backgrounds, as participation requires a smart device and internet access. Their mental health status might be underrepresented in our study. We kept the focus of our paper on the

University of Derna medical students, as medical students are a unique demographic, and Derna city was hit the hardest by the storm. A study evaluating the entire youth and young adult population of Derna would also be informative. This study was conducted at a single institution; therefore, the generalizability of its findings to other universities in Libya may be limited. Finally, longitudinal and follow-up studies are needed to assess the lasting impact of Storm Daniel on medical students.

## Conclusions

Medical students at the University of Derna reported high levels of anxiety, depression and suicidal ideation following Storm Daniel. Internal displacement was significantly associated with both anxiety and depression, and female gender was significantly associated with anxiety in the multivariate analysis.

There is a huge need for targeted interventions, accessible mental health counseling services, and a network of support for University of Derna medical students to help them cope with the aftermath of Storm Daniel, with the aim of improving their mental health and overall well-being.

Strengthening ethical practices in future disaster research, including embedding psychological support pathways, can further enhance the credibility and humanitarian value of such work.

## List of abbreviations

GAD-7 Generalized Anxiety Disorder-7
IQR inter-quartile Range
PHQ-9 Patient Health Questionnaire-9
SD standard deviation
$\chi^2$ chi-square test

**Open peer review.** To view the open peer review materials for this article, please visit http://doi.org/10.1017/gmh.2025.10039.

**Supplementary material.** The supplementary material for this article can be found at http://doi.org/10.1017/gmh.2025.10039.

**Data availability statement.** Data are available upon reasonable request from the authors.

**Acknowledgments.** The authors would like to extend their thanks to the Libyan Red Crescent team and the students of Derna Medical School for helping them with data distribution and collection.

**Author contribution.** RHS and MSB wrote the full manuscript, designed the tables and conducted the data analysis and interpretation. RHS took part in submitting the article for ethical clearance and other administrative tasks. RHS designed the questionnaire and reached out to Darna Medical School and the Libyan Red Crescent team, and distributed the questionnaire. All authors participated in conceptualization, design and analysis. All authors refined, proofread and double-checked the numbers in the final draft. All authors have read and approved the final draft for submission.

**Competing interests.** The authors declare none.

**Ethics statement.** The study was submitted to the Research Ethics Committee of the Libyan International Medical University in Benghazi, Libya. It subsequently received approval with the project number MHS-2-O-00197 and the certificate reference number MDC-2024-00160. The study was conducted in accordance with the ethical standards set by the Declaration of Helsinki. Participation was voluntary, and informed consent was obtained beforehand. The nature of the study and how we will use the information they provide were explained to participants before they took part in the survey. All responses remained anonymous, identified only by a corresponding ID number, and were securely protected with a password, accessible only to the authors.

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
