## [Reviewer Report]

This is a well-written research paper. The methodology is clearly articulated and thoroughly analysed, and the results section is informative and insightful.

Overall, the quality of the work is commendable, and I believe this research deserves to be published.

---

## [Reviewer Report]

Thank you for the opportunity to review your manuscript. It contains valuable insights on an under-researched area. Previous revisions have significantly improved the manuscript. With some minor revisions, I would recommend it for publication. Below are some of my comments to improve the conciseness and clarity of your research with a section-by-section breakdown:

1. Introduction:

The transition between the discussion of natural disasters and Libya’s recovery from the aftermath of the armed conflict could be strengthened. The transition to medical students being a unique demographic from the previous points could be strengthened as well. I also recommend including a brief explanation as to why you chose those specific study tools over others. It would also be helpful to explicitly state the inclusion criteria for participants in the study.

2. Discussion:

Overall, the organization of the discussion section felt relatively staggered. 205-207 could be better suited for the introduction section. 298-302 (touching on university response to loss of students) may be better suited for the introduction. Furthermore, the transition to 307-320 (policy and intervention recommendations) could be improved upon. 321-324 (the need for longitudinal studies) could benefit from a brief discussion on areas of further exploration, or these points could be integrated into the limitations section.

In addition, it would be helpful to discuss whether your findings are consistent with other studies similar in nature. While similar studies were mentioned, this section would benefit from an additional explanatory sentence connecting your findings to theirs.

The discussion of psychological models of PTSD and Crisis Theory is very interesting. Those sections would benefit from a more in-depth connection to the findings from your study.

While highlighted in the introductory sentence, the discussion section could benefit from a more in-depth analysis of how the armed conflict, combined with natural disasters, contributed to the psychological impact on medical students. For example, drawing on cases involving armed conflict and natural disasters impacting mental health outcomes, such as Syria (which was mentioned in the introduction, but the parallels could be clarified). It was suggested in the introduction that armed conflict could exacerbate disaster vulnerability, leading to weakened health infrastructure and poor mental health outcomes. This seemed like an interesting observation that could either be expanded upon or an area for further research.

In terms of the comparisons with findings from similar studies in other regions (i.e. the Middle East), also specify what makes Libya and Storm Daniel unique from those cases. This point could also be an area of further exploration for longitudinal studies or future research.

In terms of the recommendations for interventions, the discussion could benefit from touching upon questions such as: 1) What barriers and challenges would need to be overcome? and 2) What are the current legal, health policy and frameworks in place that require improvements based on your findings?

I encourage you to consider the recommendations to strengthen your manuscript. Overall, I see great potential and am very appreciative of your important work.

---

## [Reviewer Report]

This is an interesting study that addresses the acute psychological effects of a natural disaster in a politically fragile and under-resourced setting. The authors offer original data from a highly affected population. A few areas of concern.

I would like to know if factors like exposure severity, loss of family/friends, prior trauma, or social support were considered in the study. This will be very important for the relevance of the study. These omitted variables may have stronger explanatory power for psychological distress than demographic factors alone.

The study currently has low explanatory power. The logistic regression models for both anxiety and depression explained only ~10–13% of variance (Nagelkerke R²), indicating that most predictors of mental health outcomes remain unmeasured.

The categorization in the analysis needs to be reassessed. Income brackets (e.g., <75 LYD, 76–150 LYD, >150 LYD) may lack real-world interpretability or purchasing power context, and the meaning of “sufficient income” could vary.

Some variables were significant in univariate analysis but not in logistic regression, suggesting potential multicollinearity or inadequate adjustment. Consider including effect sizes or confidence intervals in the regression results within the main text, not only in tables.

There are several typographical errors in the manuscript. Also, there was the inconsistency in tense. Occasionally the manuscript flips between past and present tenses. Standardize to past tense for completed research.

Table 6 is mislabeled (“Logistic regression analysis of the effects of variables on the likelihood of anxiety”) but discusses depression.

Some repetitive phrasing and awkward sentence structures appear throughout (e.g., “The storm may have compounded these difficulties...” appears redundant given earlier sentences). A language edit would enhance readability.

---

## [Reviewer Report]

Thank you for the opportunity to review the paper. There has been significant improvements since the last revision. I’ve provided some minor recommendations for revision below.

On line 78 describing the selection criteria why GAD-7 and PHQ-9, the term “good psychometric properties” was used. What does “good” mean? Use more quantifiable terminology and criterion such as reliability, validity, standardized administration, and normative data. While these psychometric properties are implied in subsequent sentences, they should be explicitly stated here for clarity and cohesion.

The discussion of ethics should be included in both the main body of the paper and the ethical considerations section at the end. It would be helpful for the discussion of ethics, informed consent, and confidentiality to be interwoven throughout the paper. The paper could benefit from an explanation on how the research conducted adheres to ethical guidelines, outlining the consent process and how it was obtained from participants, and the measures taken to protect participants’ identities and ensure safe storage and handling of data. In addition to potential strengths and limitations of the approaches taken. This can be included in the methods section to describe the processes and measures used, in the discussion section as a reflection on challenges encountered during the research and steps taken to address them, and in the conclusion to reinforce the importance of ethical research—which further enhances the credibility of the findings.

There are some minor grammatical and spelling errors in the discussion section that could benefit from further edits. Overall, this paper has great potential.